# Controlling reaction pathways of selective C–O bond cleavage of glycerol

Weiming Wan[1], Salai C. Ammal[2], Zhexi Lin[1], Kyung-Eun You[2], Andreas Heyden[2] & Jingguang G. Chen[1]

The selective hydrodeoxygenation (HDO) reaction is desirable to convert glycerol into various value-added products by breaking different numbers of C–O bonds while maintaining C–C bonds. Here we combine experimental and density functional theory (DFT) results to reveal that the Cu modifier can significantly reduce the oxophilicity of the molybdenum carbide ($Mo_2C$) surface and change the product distribution. The $Mo_2C$ surface is active for breaking all C–O bonds to produce propylene. As the Cu coverage increases to 0.5 monolayer (ML), the $Cu/Mo_2C$ surface shows activity towards breaking two C–O bonds and forming ally-alcohol and propanal. As the Cu coverage further increases, the $Cu/Mo_2C$ surface cleaves one C–O bond to form acetol. DFT calculations reveal that the $Mo_2C$ surface, Cu-Mo interface, and Cu surface are distinct sites for the production of propylene, ally-alcohol, and acetol, respectively. This study explores the feasibility of tuning the glycerol HDO selectivity by modifying the surface oxophilicity.

[1] Department of Chemical Engineering, Columbia University, New York, NY 10027, USA. [2] Department of Chemical Engineering, University of South Carolina, Columbia, SC 29208, USA. These authors contributed equally: Weiming Wan, Salai C. Ammal. Correspondence and requests for materials should be addressed to A.H. (email: heyden@cec.sc.edu) or to J.G.C. (email: jgchen@columbia.edu)

With limited reserves of fossil fuel, the utilization of renewable energy resources becomes an important research topic. Biodiesel is a renewable energy source, which is produced by the transesterification of triglycerides from biomass feedstocks, such as vegetable oil and animal fats.[1] Glycerol is the main by-product of the biodiesel production; for every 100 kg of biodiesel produced, 10 kg of glycerol is formed.[2] With increasing annual production of biodiesel,[3] glycerol becomes abundant and has been considered as one of the top twelve building-block chemicals derived from biomass that can be upgraded to value-added products.[4] Therefore, a large-scale commercial process to upgrade glycerol into valuable products should create extra incentives and further promote the biodiesel production. From a chemical point of view, glycerol contains multiple hydroxy groups, and it can be used as a probe molecule to help understand the reaction mechanisms of polyols, such as glucose, fructose, or xylose.

The HDO reaction, which selectively breaks the C–O bond and preserves the C–C bond, is an important pathway to upgrade glycerol. The products of glycerol from selective HDO can be classified by the number of C–O bonds cleaved. With one C–O bond cleavage, glycerol can be converted to 1,2-propanediol on Cu-based catalysts[5,6] or to 1,3-propanediol on $M/MO_x/SiO_2$[7,8] catalysts, such as $Pt/ReO_x$ and $Pt/WO_x$. With two C–O bonds cleaved, glycerol can be converted to allyl alcohol on $K/Al_2O_3–ZrO_2–FeO_x$[9] and propanol on $Ni/Al_2O_3$[10] or $Pt/H_4SiW_{12}O_{40}$[11]. With the cleavage of all C–O bonds, glycerol can be converted to propylene and propane on Mo-based catalysts[12–14]. The above studies have identified catalysts with high activity and selectivity toward the target products, and many mechanisms and reaction pathways have been hypothesized based on reactor evaluation. However, a fundamental explanation for the selectivity toward breaking different number and type of the C–O bonds is still lacking. Such an understanding would provide guidance on how to modify the catalyst surface and tune the HDO selectivity of glycerol and other polyol molecules.

Here, we explore the possibility of tuning the HDO selectivity by manipulating the surface oxophilicity. The $Mo_2C$ surface has a high oxygen binding energy, and it has shown HDO activities for various oxygenates, such as ethylene glycol[15], furfural[16], and propanol[17]. However, the $Mo_2C$ surface breaks all C–O bonds of polyols, which makes it only promising for alkane/alkene formation, and the high oxygen binding energy impedes the facile removal of surface oxygen. To selectively break the C–O bonds in polyols and avoid C–C bond breaking, a Cu metal modifier, which has low carbon/oxygen binding energies, has been selected in the current study to tune the HDO selectivity of the $Mo_2C$ surface.

In this work, surface science experiments and theoretical calculations were combined to explore the feasibility of tuning the selectivity for the glycerol HDO by using Cu to modify the $Mo_2C$ surface. Temperature Programmed Desorption (TPD) and High-resolution Electron Energy Loss Spectroscopy (HREELS) measurements identified the gas-phase products and surface intermediates, respectively. DFT calculations revealed the most preferred reaction pathways on the Cu surface, Cu-$Mo_2C$ interface, and $Mo_2C$ surface by comparing the reaction energies and barriers of each elementary step. The results demonstrate for the first time that the Cu coverage can significantly affect the oxophilicity of the Cu/$Mo_2C$ surface and consequently tune the HDO selectivity of glycerol.

## Results

**Gas-phase products from TPD measurements.** Four products were observed in the TPD experiments from Cu/$Mo_2C$/Mo(110)

**Fig. 1** Reaction pathways for the selective hydrodeoxygenation of glycerol. Three reaction pathways were observed in the TPD experiments. Glycerol were converted to **1** propylene by breaking three C–O bonds, **2** allyl alcohol/propanal by breaking two C–O bonds, and **3** acetol by breaking one C–O bond

surfaces, propylene, allyl alcohol, propanal, and acetol. Three reaction pathways of glycerol were classified based on the number of C–O bonds cleavage, as shown in Fig. 1.

Atomic oxygen and atomic hydrogen were formed in all reactions, which can recombinatively desorb in the form of water. In all TPD experiments, 4 Langmuir hydrogen was pre-dosed on the surface, which provided extra atomic hydrogen to remove the surface oxygen. This process removes the surface oxygen and creates active sites to carry out HDO reactions. Only trace amount of atomic carbon was deposited after each TPD experiment, which was observed in AES measurements.

Figure 2 shows the desorption of four products from the hydrogen pre-dosed Cu/$Mo_2C$ surfaces with different Cu coverages. A sharp peak at 260 K was observed in all spectra, which was from the desorption of unreacted glycerol. The main cracking pattern of glycerol is at $m/z = 61$, and the spectra are shown in Supplementary Fig. 2a. The desorption of propylene, with a main cracking pattern of $m/z = 39$, is shown in Fig. 2a. The spectra of another main cracking pattern of propylene, $m/z = 41$, are shown in Supplementary Fig. 2b. The broad peak with an on-set temperature of 300 K and a peak temperature of 411 K was from propylene desorption, with the $Mo_2C$/Mo(110) surface showing the largest desorption peak of propylene, and its peak area decreasing with increasing Cu coverage. Another peak at 360 K was observed in the spectra of surfaces with high Cu coverages. This peak was from the desorption of acetol, which has a minor cracking pattern at the mass $m/z = 39$. The results in Fig. 2a indicate that the $Mo_2C$/Mo(110) surface was active for propylene formation, and Cu modification reduced the ability for C–O bond breaking.

Figure 2b shows the spectra of $m/z = 43$, which is the main cracking patterns of acetol. The spectra of another main cracking pattern, $m/z = 31$, are shown in Supplementary Fig. 2c. The sharp peak at ~365 K was from acetol desorption, and its peak area increased with increasing Cu coverage. This result suggested that high Cu covered $Mo_2C$ surfaces showed higher activity to break only one C–O bond of glycerol to form acetol. Other dehydration products such as acrolein were not observed.

Figure 2c and d shows the spectra of $m/z = 57$ and 58, which are the main cracking patterns of allyl alcohol and propanal, respectively. The peak areas of both peaks reached their maximum on 0.6 ML Cu/$Mo_2C$/Mo(110). Since both $Mo_2C$ and Cu surfaces showed low activity for this reaction, the allyl alcohol and propanal were most likely formed at the Cu-$Mo_2C$ interface. Ally-alcohol showed a desorption peak at 347 K. Propanal had a desorption peak at 389 K on Cu-lean surfaces, and the peak shifted to 375 K on Cu-rich surface. The production of both allyl alcohol and propanal were reaction limited, which was revealed in the HREELS experiments (Fig. 3). The higher desorption temperature of propanal suggested that the propanal formation

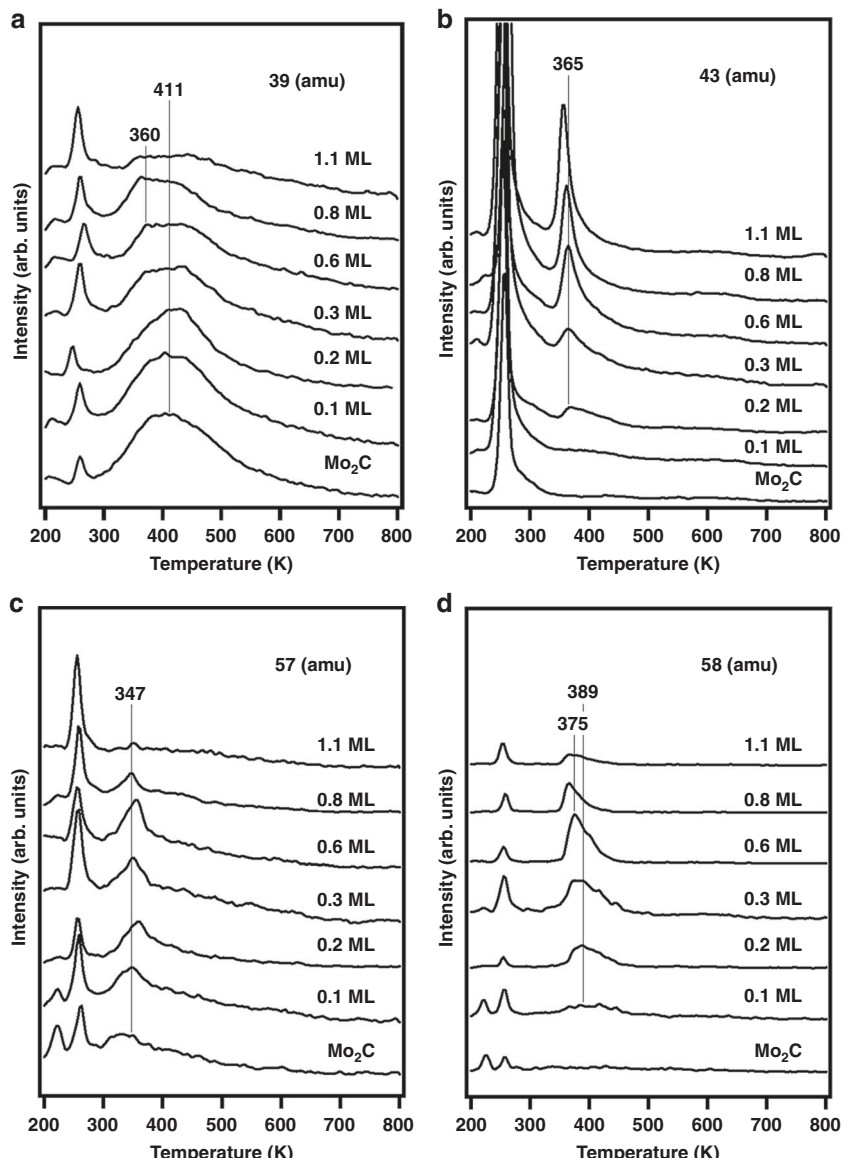

**Fig. 2** Gas-phase products of the TPD experiments. In total, 4 L glycerol was exposed on hydrogen pre-dosed Cu/Mo$_2$C surfaces with different Cu coverages. Spectra of **a** propylene ($m/z = 39$), **b** acetol ($m/z = 43$), **c** ally-alcohol ($m/z = 57$), and **d** propanal ($m/z = 58$)

had a slightly higher activation barrier than that of allyl alcohol. On Cu-rich surfaces, the peak area of propanal was larger than allyl alcohol, while the trend was reversed on Cu-lean surfaces. These results suggested that Cu might play a role to convert allyl alcohol to propanal through the isomerization reaction. This hypothesis is consistent with a previous study that reported Cu (110) as being active for converting allyl alcohol to propanal[18]. On the Mo$_2$C/Mo(110) and 0.1 ML Cu/Mo$_2$C/Mo(110) surfaces, a small amount of allyl alcohol and propanal were observed, which can be from defected Mo$_2$C sites. As Cu coverage increased, these sites were occupied by Cu.

The peak area and mass spectrometry sensitivity factor of each product were used to quantify the yields of the three reactions from Fig. 1. The quantification results were summarized in Supplementary Table 2, and the yield of Reaction 2 was the sum of the yields of propanal and allyl alcohol. As the Cu coverage increased from 0 to 1.1 ML, the yield of Reaction 1 decreased from 0.068 to 0.013 and that of Reaction 3 increased from 0 to 0.062. At the 0.6 ML Cu coverage, the Reaction 2 reached its maximum yield of 0.041.

The detection of water desorption indicated the removal of the atomic oxygen formed in the glycerol HDO reactions, as shown for various Cu/Mo$_2$C/Mo(110) surfaces in Supplementary Fig. 3a. A broad peak at 569 K was observed on the Mo$_2$C surface, and a sharp peak at 365 K was observed on the Cu-terminated surface. These results suggested that the oxygen removal from the Mo$_2$C surface was more difficult than that from the Cu-modified surfaces, which should be due to the strong oxophilicity of the Mo$_2$C surface. DFT calculations further confirmed that for Mo$_2$C surface sites the desorption of H$_2$O is more endothermic, and the activation barrier for the oxygen removal process is nearly 1 eV higher in the absence of gas-phase H$_2$ than for the Cu/Mo$_2$C interface sites and Cu surface sites (Supplementary Table 1). The hydrogen coverage effect was studied (Supplementary Fig. 4) and the pre-dosed amount did not affect the activities of the three distinguish sites. To study the role of Cu modifier in catalyst regeneration, sequential TPD experiments were performed under UHV conditions (Supplementary Fig. 5), suggesting that the Mo$_2$C surface and Cu-Mo$_2$C interface are less stable than the Cu surface. This is related to the strong oxophilicity of the Mo site

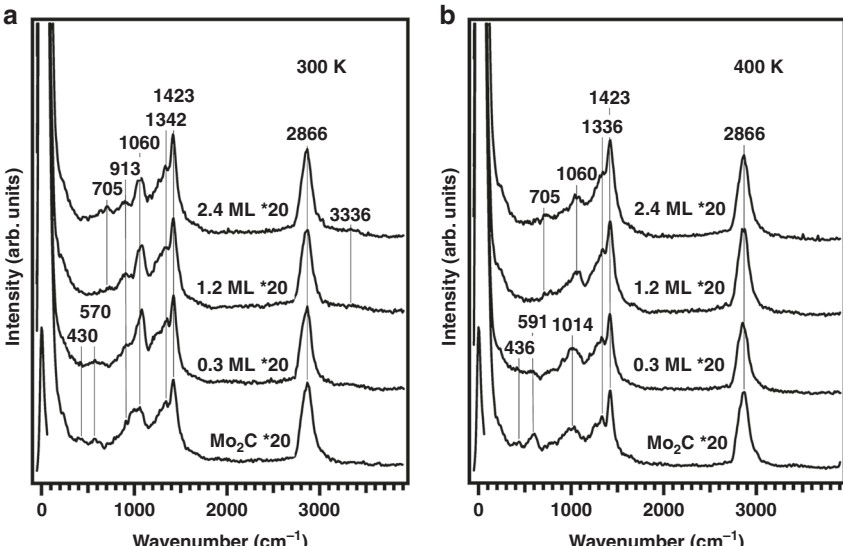

**Fig. 3** HREELS measurements of the surface intermediates. Glycerol was exposed on H₂ pre-dosed Mo₂C/Mo(110), 0.3 ML Cu/Mo₂C/Mo(110), 1.2 ML Cu/Mo₂C/Mo(110), and 2.4 ML Cu/Mo₂C/Mo(110). Spectra at **a** 300 K and **b** 400 K

that prevents oxygen removal under UHV conditions. However, in a real catalytic HDO process with a high H₂ pressure the gas-phase H₂ should help remove surface oxygen (Supplementary Discussion), as observed in a previous study.[17]

Overall the TPD experiments revealed three types of pathways for the HDO reaction of glycerol, with the yields of the corresponding products showing distinct trends with increasing Cu coverage. As the Cu coverage increases, the surface prefers to break fewer C–O bonds of glycerol. Therefore, the selectivity of the glycerol HDO reaction can be tuned by changing the Cu coverage on the Mo₂C surface.

**HREELS results of glycerol on Mo₂C and Cu/Mo₂C surfaces.** HREELS experiments were performed to identify surface reaction intermediates on four surfaces: Mo₂C/Mo(110), 0.3 ML Cu/ Mo₂C/Mo(110), 1.2 ML Cu/Mo₂C/Mo(110), and 2.4 ML Cu/ Mo₂C/Mo(110). The spectra of molecularly adsorbed glycerol on the four surfaces were shown in Supplementary Fig. 6a. Molecular adsorption was indicated by the similarities among the four spectra, as well as the observation of the $\tau$ (OH) mode at 671 cm$^{-1}$ and the $\nu$(OH) mode at 3248 cm$^{-1}$. The detailed assignments of each vibrational mode are summarized in Supplementary Table 3.

Unreacted glycerol desorbed at 280 K (shown in the TPD results in Fig. 2), and Fig. 3a shows the spectra of the chemisorbed glycerol. Comparing the spectra of four surfaces, from 0 to 2.4 ML Cu coverage, the Mo₂C/Mo(110) surface shows a strong oxophilicity indicated by the red-shifting of the $\nu$(CO) mode (1060 cm$^{-1}$) and the appearance of the two vibrational modes (430 and 570 cm$^{-1}$) from the Mo–O bond. Both $\tau$ (OH) (705 cm$^{-1}$) and $\nu$(OH) (3336 cm$^{-1}$) modes were not observed on Mo₂C/Mo(110), suggesting that the chemisorbed glycerol underwent dissociative adsorption by breaking all O–H bonds and forming Mo–O bonds. As the Cu coverage increased to 0.3 ML, the $\nu$(CO) mode was not red-shifted and the intensities of the Mo–O vibrational modes were decreased suggesting the surface oxophilicity was reduced. A very small peak from the $\nu$(OH) (3336 cm$^{-1}$) mode was observed, suggesting the chemisorbed glycerol retained some of the O–H bonds. On the 1.2 ML Cu/ Mo₂C/Mo(110) surface, the vibrational modes from the Mo–O/ Cu–O bond were not observed, and the $\nu$(CO) mode (1060 cm$^{-1}$) was not red-shifted, which suggested an even

weaker interaction between the Cu surface and glycerol. The detection of the $\tau$ (OH) mode (705 cm$^{-1}$) and the weak $\nu$(OH) mode (3336 cm$^{-1}$) indicated the existence of O–H bonds in the chemisorbed glycerol. The spectrum of 2.4 ML Cu/ Mo₂C/Mo (110) is similar to that on 1.2 ML Cu/ Mo₂C/Mo(110), showing that the surface interaction of glycerol remained the same as the Cu coverage exceeded 1 ML.

Figure 3b shows the spectra of glycerol after annealing the surface to 400 K. On Mo₂C/Mo(110), the $\nu$(CO) mode further red-shifted, the intensity of the $\nu$(CO) mode was reduced and the intensities of $\delta$(Mo–O) and $\nu$(Mo–O) increased. This spectrum indicated that more C–O bonds were cleaved and more Mo–O bonds were formed. On 0.3 ML Cu/ Mo₂C/Mo(110), the $\nu$(CO) red-shifting was observed, indicating this surface could also weaken the C–O bonds. Compared with Mo₂C/Mo(110), the Mo–O modes were less intense and the $\nu$(CO) peak intensity was slightly higher, indicating that the Cu modifier reduced the activity in C–O bond cleavage. On 1.2 ML Cu/Mo₂C/Mo(110) and 2.4 Cu/Mo₂C/Mo(110), the $\nu$(CO) mode was not red-shifted. The intensity of $\nu$(CO) was slightly reduced due to the desorption of acetol (Fig. 2b) at 365 K.

As the surface temperature increased to 500 K (Supplementary Fig. 6b), the $\nu$(CO) mode on Mo₂C/Mo(110) and 0.3 ML Cu/ Mo₂C/Mo(110) almost vanished, and the Mo–O modes became significant, suggesting the cleavage of all the C–O bonds. On Cu-terminated surfaces, the $\nu$(CO) intensity was slightly reduced and the Cu–O mode were not observed due to the removal of the surface oxygen by H₂O formation.

In summary, HREELS results were consistent with the TPD conclusions that the presence of a Cu modifier on Mo₂C reduced the oxophilicity of the surface. The oxophilicity was reflected by the red-shift of the $\nu$(CO) mode and the increase in intensities of the Mo–O modes. The oxophilicity of the Cu/Mo₂C/Mo(110) surfaces decreased by increasing the Cu coverage. The weak oxophilicity of the Cu-terminated surface suppressed the C–O bond cleavage, but it also helped the removal of surface oxygen, which was consistent with the disappearance of the Mo–O/Cu–O modes on the Cu-terminated surface. The $\nu$(C=O) and $\nu$(C=C) modes were not observed on any surfaces, suggesting that the desorption of acetol and allyl alcohol was reaction limited, i.e., the products desorbed from the surface as soon as they were produced.

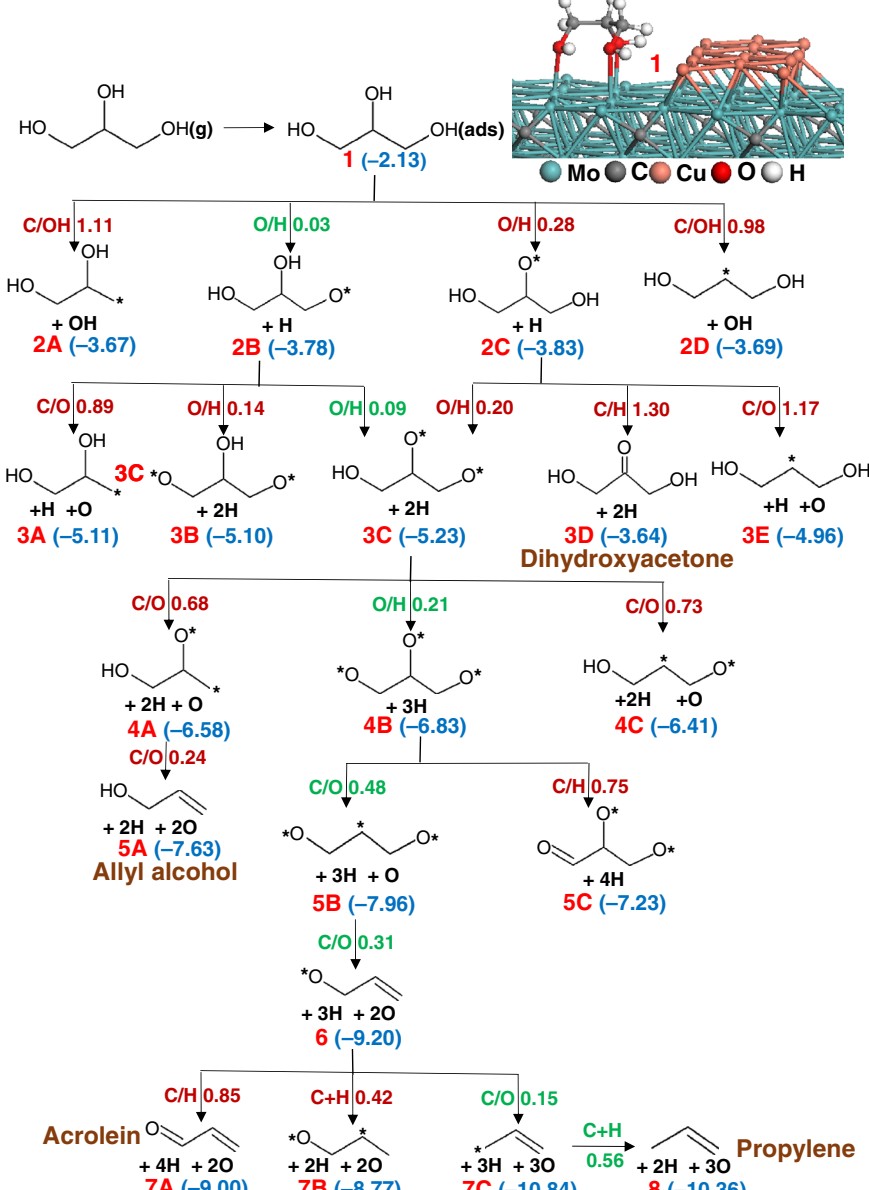

**Fig. 4** Schematic of the reaction pathways for glycerol decomposition on the Mo sites of the Cu/Mo$_2$C (0001) catalyst model. The activation barriers (in eV) of the elementary reactions are shown next to the arrows, and the numbers shown in parentheses are the energies of the intermediates relative to gas-phase glycerol and the initial catalyst model. * represents an undercoordinated atom and the most favorable pathway is shown in green

**Theoretical calculation of glycerol HDO on Cu/Mo$_2$C site models.** In this section, the deoxygenation mechanism of glycerol is investigated from first principles on three active sites of our Cu/Mo$_2$C catalyst model: a Mo$_2$C site, a Cu site, and a Cu/Mo$_2$C interface site. The deoxygenation mechanism of complex polyols, such as glycerol, to all theoretically possible reaction products, involves a very large number of elementary reactions that is currently beyond the capability of DFT. While a number of theoretical studies have examined various decomposition and hydrogenolysis pathways of glycerol on transition metal surfaces, semiempirical correlations had to be used in addition to periodic DFT calculations, and only the closed-packed surfaces of transition metal catalysts could be investigated.[19–25] Here, the focus is on understanding the experimentally observed selectivity of the different active sites for glycerol leading to four different products, propylene (CH$_2$=CH–CH$_3$), acetol (CH$_3$–CO–CH$_2$OH), allyl alcohol (CH$_2$=CH–CH$_2$OH), and propanal (CH$_3$–CH$_2$–CHO) on

Cu/Mo$_2$C catalysts. Thus, we focussed on investigating only the key competing reaction pathways leading to these four products. Specifically, the elementary reactions considered here include all three O–H bond cleavages, all three C–O bond cleavages, one C–H bond cleavage at a terminal and central carbon atom, and relevant hydrogenation steps to the observed products. C–C bond cleavage was not investigated considering that it was not observed experimentally.

**Deoxygenation of glycerol on the Mo sites.** Figure 4 illustrates various elementary reaction pathways for the deoxygenation of glycerol to propylene on the Mo sites of our Cu/Mo$_2$C catalyst model. All reaction energies and activation barriers are zero-point energy corrected and are summarized in the Supplementary Table 4. The optimized structures of corresponding intermediates and transition states are depicted in Supplementary Figs. 7, 8. Glycerol adsorbs strongly on the Mo$_2$C (0001) surface with all

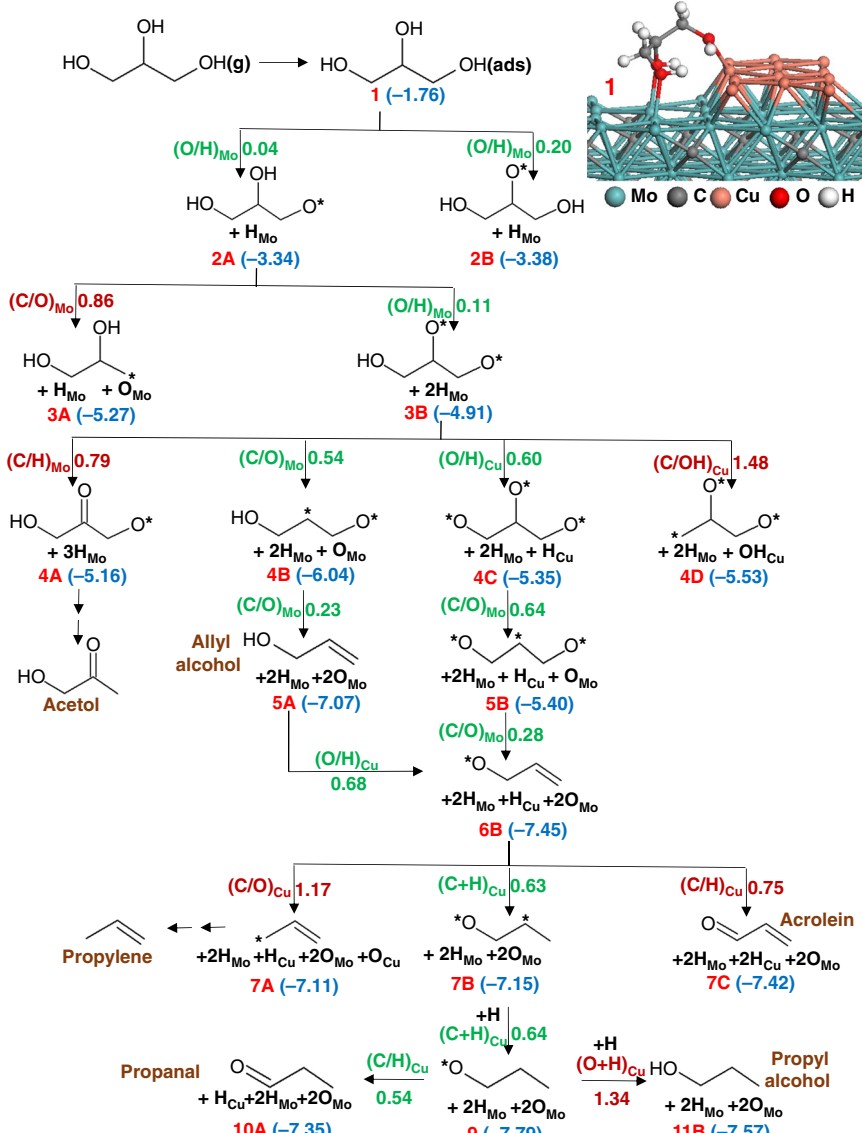

**Fig. 5** Schematic of the reaction pathways for glycerol decomposition at interface sites of the Cu/Mo$_2$C (0001) catalyst model. The activation barriers (in eV) of the elementary reactions are shown next to the arrows, and the numbers shown in parentheses are the energies of the intermediates relative to gas-phase glycerol and the initial catalyst model. * represents the undercoordinated atom and the most favorable pathways are shown in green

three oxygen atoms interacting with three Mo sites located above the second layer carbon atoms with a binding energy of −2.12 eV (see the inset of Fig. 4). The relatively stronger binding energy of glycerol compared to the reported binding energy of ethylene glycol on Mo$_2$C (0001) surface (−1.07 eV calculated with PBE functional)[15] is due to binding of an additional oxygen atom to a Mo site and from the inclusion of vdW interactions in our computed adsorption energy. It has been shown that the inclusion of vdW interactions is essential to improve the long-range interaction of glycerol with the surface. For example, the adsorption of glycerol on flat and defected Pt (111) substrates can be stabilized by nearly 1 eV by the addition of the D3 vdW correction to the PBE functional.[26] Comparison of the activation barriers of initial O–H and C–OH bond cleavages of glycerol suggest that the O–H scissions are preferred and possess very low activation barriers (0.03–0.28 eV). It should be noted here that although some of the reactant or product states in the elementary reactions shown in Supplementary Tables 4–6 are represented as consisting of multiple species, in the energy calculations of each reactant or product state they constitute one discrete state, i.e., the dissociated atoms/

fragments stay on the surface and lateral interactions between these species are considered at the DFT level.

In the second step, O–H, C–O, and C–H bond cleavages from the two alkoxide species (CH$_2$OH–CHOH–CH$_2$O* and CH$_2$OH–CHO–CH$_2$OH*) were examined and we observed that the O–H scissions are again highly favorable. Although the C–O scissions are thermodynamically equally favorable, the barriers for these reactions (0.9–1.2 eV) are significantly higher than the O–H cleavage barriers (0.1–0.2 eV). In the next step, the third O–H scission is again favorable ($E_{ZPE}^{act} = 0.21$ eV) relative to the C–O bond dissociations ($E_{ZPE}^{act} = 0.68$–0.73 eV) leading to the formation of allyl alcohol. These trends are consistent with previous reports on the ethanol decomposition on the α-Mo$_2$C (100) surface[27] and ethylene glycol decomposition on β-Mo$_2$C (0001) surface,[15] both suggesting the preferential dissociation of O–H bonds prior to any other C–O or C–H bond cleavages.

Next, the intermediate CH$_2$O–CHO–CH$_2$O* can undergo three subsequent C–O bond dissociations, first at the middle carbon ($E_{ZPE}^{act} = 0.48$ eV) followed by the two terminal carbons with activation barriers of only 0.31 and 0.15 eV, respectively. C–H

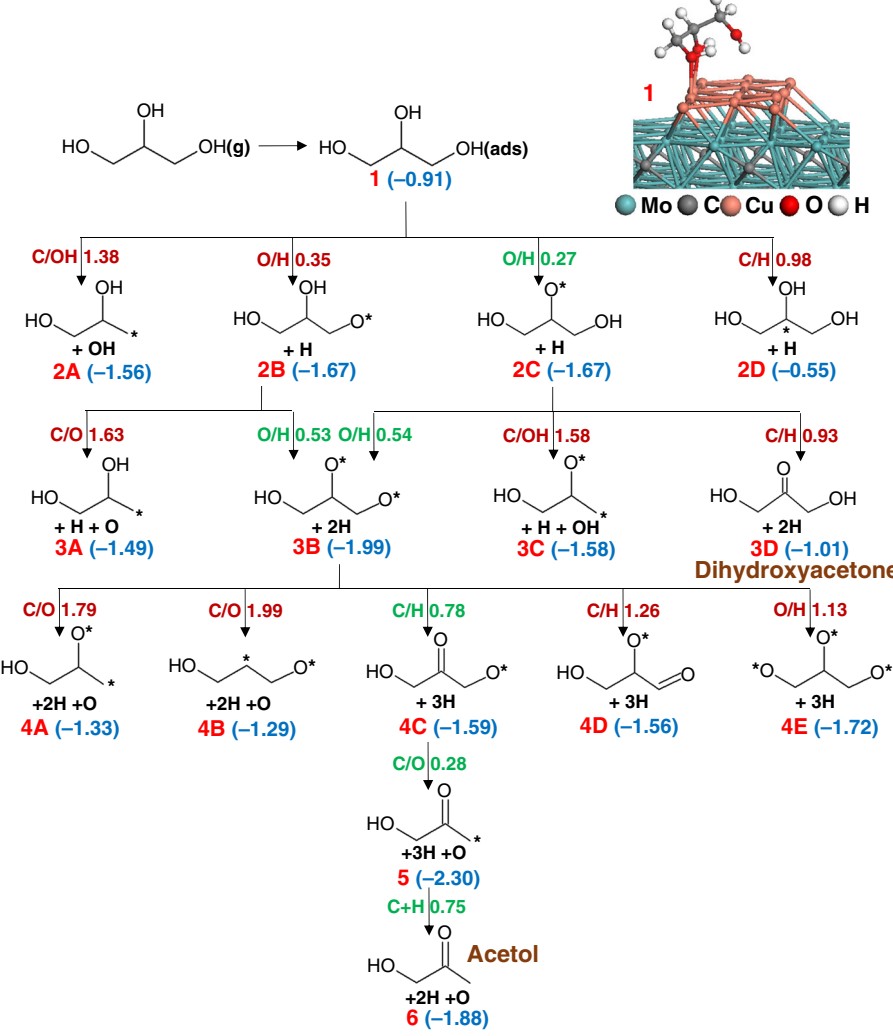

**Fig. 6** Schematic of the reaction pathways for glycerol decomposition on Cu sites of the Cu/Mo$_2$C (0001) catalyst model. The activation barriers (in eV) of the elementary reactions are shown next to the arrows, and the numbers shown in parentheses are the energies of the intermediates relative to gas-phase glycerol and the initial catalyst model. * represents the undercoordinated atom and the most favorable pathway is shown in green

bond scission and protonation reactions, after one or two C–O bond dissociations, were also examined and found to possess higher activation barriers than the subsequent C–O dissociation steps. The final step for propylene production involves the protonation at a terminal carbon of the CH$_2$–CH=CH$_2$* intermediate. The activation barriers calculated for this protonation step (0.56 eV) and the first C–O scission from the CH$_2$O–CHO–CH$_2$O* intermediate (0.48 eV) appear to be the largest barriers in the minimum energy pathway for the formation of propylene from glycerol on the Mo sites. Considering the small magnitude of these barriers, the computational results agree very well with the experimental observation that propylene is the major product on pure Mo$_2$C and on the Cu/Mo$_2$C catalysts with low Cu coverage.

**Deoxygenation of glycerol at the Cu/Mo$_2$C interface sites**. The reaction pathways examined for the glycerol deoxygenation mechanism at the interface sites of Cu/Mo$_2$C are depicted in Fig. 5. The corresponding reaction energies and optimized structures are summarized in the Supplementary Table 5 and Supplementary Figs. 9, 10. Computations predict that the structure in which two oxygen atoms of glycerol are interacting with Mo atoms and one oxygen of glycerol is interacting with a Cu

atom is the preferred adsorption mode for glycerol ($E_{ZPE}^{ads} = -1.76$ eV) at the Cu/Mo$_2$C interface (inset of Fig. 5). The two hydroxyl groups adsorbed on the Mo sites undergo rapid O–H bond scissions with an activation barrier of only 0.04 eV for the terminal –OH followed by a barrier of 0.11 eV for the middle O–H scission. Upon formation of the alkoxide intermediate CH$_2$OH–CHO–CH$_2$O*, the reaction flux can follow a third O–H bond scission on the Cu sites ($E_{ZPE}^{act} = 0.60$ eV) or a C–O bond scission on the Mo sites ($E_{ZPE}^{act} = 0.54$ eV). If the reaction flux progresses through the slightly more favorable C–O bond cleavage pathway, allyl alcohol (CH$_2$OH–CH=CH$_2$*) can be produced after a second C–O bond cleavage that requires only a minimal barrier of 0.23 eV. After each C–O bond cleavage, the oxygen atom stays on the Mo sites and the carbon moiety moves to the Cu sites and thus, the final product, allyl alcohol is adsorbed on the Cu sites. Among the five elementary reactions involved in the minimum energy pathway of allyl alcohol formation from glycerol, the first C–O bond cleavage reaction has the largest activation energy (0.54 eV) and thus, it is likely the rate-limiting process. A C–H bond scission at the middle carbon of the CH$_2$OH–CHO–CH$_2$O* intermediate, which leads to the formation of acetol (CH$_2$OH–CO–CH$_3$*), was also examined; however, the activation barrier is found to be 0.25 eV higher than that for C–O dissociation. Thus, in agreement with experimental

observations, computations predict that at the interface sites the formation of allyl alcohol is highly favorable.

The intermediate, CH₂O–CHO–CH₂O* formed after the third O–H bond dissociation can also undergo two subsequent C–O bond cleavage reactions with activation barriers of 0.64 and 0.28 eV, respectively, forming the intermediate CH₂O–CH=CH₂*. This intermediate can also be produced by the deprotonation reaction of allyl alcohol on the Cu sites with an activation barrier of 0.68 eV. A third C–O bond cleavage on the Cu sites, which can lead to the formation of propylene, was found to possess a large barrier of 1.17 eV. The formation of the other experimentally observed product, propanal (CHO–CH₂–CH₃*), can be produced from the CH₂O–CH=CH₂* intermediate via a C–H bond dissociation at the alkoxy group and a hydrogenation step of the alkene moiety. For the C–H bond dissociation of the alkoxy group, leading to adsorbed acrolein (CHO–CH=CH₂*), we compute an activation barrier of 0.75 eV. Alternatively, protonation of the terminal –CH₂ group ($E_{ZPE}^{act} = 0.63$ eV) followed by C–H dissociation at the alkoxy group ($E_{ZPE}^{act} = 0.54$ eV) is found to be the favorable pathway for the formation of propanal. Adsorption and desorption steps of H atoms on the Cu rod are not shown for clarity in Fig. 5; however, the reaction energies for these steps have been considered. These results agree with an earlier report by Brainard et al.[18] who found that allyl acohol can react with clean and oxygen-covered Cu(110) surfaces to produce propanal, acrolein and n-propyl alcohol under ultrahigh-vacuum conditions and that hydrogenation of the alkene occurs prior to C–H bond dissociation necessary for the formation of propanal. Since acrolein and propyl alcohol were not observed in the current experimental studies and the computations also suggest that the formation of these products is less favorable than the formation of propanal, it is reasonable to suggest that the Cu monolayer adsorbed on Mo₂C behaves differently than a pure Cu surface and that it is more selective for the conversion of allyl alcohol to propanal.

**Deoxygenation of glycerol on the Cu sites**. Figure 6 illustrates the reaction pathways of the glycerol deoxygenation examined solely on the Cu sites. The corresponding elementary reactions with energies and structures can be found in Supplementary Table 6 and Supplementary Figs. 11, 12. Since the preferential adsorption site for Cu on the Mo₂C surface is in the Mo hollow sites, and the distance between two neighboring hollow sites on the β-Mo₂C(0001) surface (≈ 3.0 Å) is larger than the Cu–Cu distance in bulk Cu (2.56 Å as predicted by PBE functional), the Cu–Cu bonds in the Cu rod are elongated with the distance between neighboring Cu atoms ranging from 2.6–3.2 Å. Thus, during the reaction, these Cu atoms can slightly move and create holes in the Cu rod structure which permits the dissociated H atoms to slip into these holes, forming a bond with subsurface Mo atoms as shown in the latter part of this mechanism. We found that such holes on the Cu layer can be created as long as the coverage of Cu remains below 1 ML.

Adsorptions of glycerol with one, two, and three O–Cu interactions were investigated, and we found that the most favorable structure involves two O–Cu bonds (inset of Fig. 6). The calculated adsorption energy (−0.91 eV) is larger than the reported adsorption energy on the Cu(111) surface (−0.20 eV calculated with PW91 functional)[28], which is mainly due to the inclusion of vdW corrections in the present study. In contrast, the adsorption of glycerol on the Cu rod is less favorable than the adsorption on the Mo sites and at the interface sites by 1.22 eV and 0.85 eV, respectively. Since the current TPD experiments reveal that acetol is formed only in the presence of Cu on Mo₂C and the yield of acetol formation increases with increasing Cu

coverage (Supplementary Table 2), the mechanistic study on the Cu sites focuses mainly on the formation of acetol while few elementary reactions that can lead to the formation of the other three experimentally observed products were also examined.

The four elementary reactions investigated for the first dissociation of glycerol on Cu (Fig. 6) suggest that O–H bond cleavage is preferable compared to a C–OH scission at the primary carbon or C–H scission at the secondary carbon. The activation barriers for the O–H scission reactions are higher than those on the Mo sites. The calculated barrier for the middle O–H scission ($E_{ZPE}^{act} = 0.27$ eV) is much lower than the reported barrier on Cu (111) surface (0.80 eV calculated with PW91 functional)[25] illustrating the higher activity of the Cu sites of the Cu/Mo₂C catalyst relative to Cu(111). The intermediate, CH₂OH–CHO–CH₂OH*, also seems to favor an O–H bond scission relative to C–OH and C–H bond dissociations at the primary and secondary carbons, respectively. The larger barrier calculated for the C–H scission at the secondary carbon ($E_{ZPE}^{act} = 0.93$ eV) suggests that the formation of dihydroxyacetone is not favorable. It is to be noted that the first and second O–H bond scission reactions on Cu are exothermic only by −0.76 eV and −0.32 eV, respectively, compared with the large exothermicity of >1.5 eV for each O–H bond scission on the Mo sites. Clearly, Cu possesses a much weaker affinity for O and H than Mo. Next, the third O–H bond cleavage and various C–O bond cleavages, that were found to be favorable on the Mo sites, were investigated. In addition, we studied various C–H bond cleavages. A comparison of the activation barriers of these elementary reactions suggests that a C–H scission at the secondary carbon forming the intermediate CH₂OH–CO–CH₂O* is the most favorable pathway, which can subsequently undergo a C–O bond scission at the terminal carbon with a low barrier of 0.28 eV. Although this C–O bond cleavage barrier seems to be quite low for a Cu site, it should be noted that the previous C–H cleavage reaction is endothermic by 0.40 eV which increases the effective barrier for C–O cleavage to 0.68 eV. In addition, the presence of a ketone group in the intermediate, CH₂OH–CO–CH₂O*, seems to facilitate the C–O scission at the neighboring carbon atom forming a stable enolate intermediate (CH₂OH–CO–CH₂*). The final step is the protonation of the terminal –CH₂ group which possesses a barrier of 0.75 eV and leads to the formation of acetol. In agreement with the TPD experiments, these calculations suggest that the formation of acetol is favorable only on the Cu sites because of the preferential formation of a ketone group, whereas the Mo and interface sites prefer to cleave the central C–O bond of glycerol. The correlation of computational predictions to TPD and HREELS experimental results are provided in the Supplementary Discussion. Here, we also show computational results that suggest O adatoms deposited on Mo sites in direct neighborhood of Cu sites can be regenerated by H₂ at high oxygen coverage that is likely present under practical reactor conditions (Supplementary Fig. 16).

## Discussion

Cu-modified Mo₂C surfaces contain three distinct active sites: Mo₂C, Cu, and the Cu–Mo₂C interface. The strong oxophilicity of the Mo₂C surface makes it active towards breaking all C–O bonds in glycerol to produce propylene. The Cu modifier reduces the oxophilicity of the Mo₂C surface. At moderate Cu coverage, the Cu–Mo₂C interface shows distinct activity to cleave two C–O bonds to form allyl alcohol, which can further be converted to propanal through isomerization. The high Cu coverage surface shows activity toward breaking one C–O bond and forming acetol. The results obtained from DFT calculations on the glycerol decomposition pathways at the three different active sites agree very well with the experimental observations. Competing

elementary reactions that can lead to all four products of interest were examined at different active sites, and the minimum energy pathways revealed a selectivity for propylene formation on Mo sites, acetol formation on Cu sites, and allyl alcohol together with propanal formation at the interface sites. Using the same catalyst model for the investigation of three different active sites allowed us to compare the energetics of similar reactions on different sites. Mo sites exhibit high activity and selectivity for O–H and C–O bond cleavage reactions, whereas the activation barriers for these reactions are higher on the Cu sites. Interestingly, Cu adsorbed on $Mo_2C$ still exhibits a higher activity for O–H and C–O bond cleavage reactions compared with the Cu (111) surface. Although the reactions occur easily on the Mo sites, the stronger affinity of Mo toward O and H can leave the Mo sites poisoned with these adatoms whereas the products easily desorb from the Cu sites. Interestingly, computational results suggest that the O adatoms deposited on Mo sites in direct neighborhood of Cu sites can be regenerated by $H_2$ at high oxygen coverage likely present under reactor conditions.

Overall, this work demonstrates the feasibility of partially modifying the $Mo_2C$ surface with Cu to generate distinct active sites with different oxygen binding energies on the surface, which can be used to control the number of cleaved C–O bonds in the glycerol hydrodeoxygenation.

## Methods

**Experimental methods.** The TPD experiments were performed in an ultrahigh vacuum (UHV) chamber with a $5 \times 10^{-10}$ Torr base pressure. The UHV chamber was equipped with an Auger Electron Spectrometer (AES), metal evaporation source, sputter gun, and quadrupole mass spectrometer (UTI100). In a typical TPD experiment, the crystal was cooled down to 200 K and exposed to 4 Langmuirs (L; $1 \text{ L} = 10^{-6}$ Torr.s) $H_2$ and 10 L glycerol. Then the crystal was further cooled down to 100 K and heated to 800 K at a linear rate of 3 K/s. The gas-phase products were measured with the mass spectrometer.

The HREELS experiments were conducted in another UHV chamber equipped with TPD and AES capabilities. The HREELS spectra were acquired with a primary beam energy of 6 eV. Angles of incidence and reflection were 60° with respect to the surface normal. Count rates in the elastic peak were typically between $1 \times 10^4$ and $3 \times 10^4$ counts/s, and the spectral resolution was between 30 and 50 cm$^{-1}$ full width at half maximum. 5 L $H_2$ was pre-dosed on each surface, and 4 L glycerol was adsorbed at a surface temperature of 200 K. In each measurement, the surface was flashed to the indicated temperature at 3 K/s, maintained at the indicated temperature for 1 min and cooled down to 120 K before the HREELS spectrum was recorded.

Glycerol (Fisher Scientific, 99%) was transferred into a glass sample cylinder and purified using repeated heat-pump-cool cycles. All the other gases, oxygen, hydrogen, neon, ethylene, and carbon monoxide, were of research purity and used without further purification. The purity of all the reagents was verified before experiments using in situ mass spectrometry. The glycerol was preheated to 400 K before being dosed into the UHV system.

The Mo(110) single crystal (Princeton Scientific, 99.99%) was 1-mm thick with a 10-mm diameter and was oriented to within 0.5 degrees. The crystal was spot welded directly to two tantalum posts that served as electronic contacts for resistive heating, as well as thermal contacts for cooling with liquid nitrogen. The Mo(110) single crystal was cleaned by Ne$^+$ sputtering at 300 K followed by annealing to 1200 K. Oxygen treatment was performed following the Ne$^+$ sputtering to remove the surface carbon. In the oxygen treatment, the single crystal was heated to 1000 K and exposed to 100 L of oxygen, followed by annealing to 1200 K. The oxygen treatment was repeated until negligible amounts of carbon and oxygen were observed on Mo(110). The carbide-modified Mo(110) was synthesized by ethylene treatment on cleaned Mo(110). In each ethylene treatment, Mo(110) was held at 600 K and exposed to 3 L of ethylene, followed by annealing to 1200 K. Cycles of ethylene treatment were used until the C(275 eV)/Mo(186 eV) AES ratio reached to an atomic Mo/C ratio of ~2:1. The procedure for the $Mo_2C$ surface preparation was similar to the previous study.[29]

The Cu-modified $Mo_2C$/Mo(110) surface was prepared by evaporative deposition of Cu atoms onto $Mo_2C$/Mo(110). The crystal was held at 300 K during deposition. The metal source consisted of a tungsten filament with a high purity Cu wire (Alfa Aesar, 99.99 + %) wrapped around it, which was mounted within a tantalum enclosure. The Cu coverage on the $Mo_2C$/Mo(110) surface was controlled by the deposition time and quantified using the AES peak-to-peak heights. The relationship between AES intensities and the deposition time was measured and plotted in Supplementary Fig. 1a. The intensities of both Cu(64 eV) and Mo(186 eV) show distinct breaks at 7 min, and the AES ratio of Cu(64 eV)/Mo(186 eV) at 7 min was 0.55, which was close to the theoretical result of one monolayer Cu on

$Mo_2C$[30]. This result suggested that the Cu deposition on $Mo_2C$/Mo(110) followed the epitaxial growth mechanism for the first layer and one ML of Cu was formed after 7 min of Cu deposition. The Supplementary Fig. 1b shows the thermal stability of ~1 ML Cu on $Mo_2C$ surface. The Cu/Mo AES ratio is within 10% of the initial value at temperatures below 600 K, suggesting that the Cu overlayer remains relatively stable in the temperature range of the HDO reactions.

**Computational methods.** DFT calculations were carried out on a Cu/$Mo_2C$ catalyst model to understand the experimentally observed selectivity pattern for the HDO of glycerol and to identify likely active sites. All calculations were performed using the Vienna ab initio simulation package (VASP).[31–34] The projector augmented wave (PAW)[28,35] method was used to describe the electron–ion interaction, whereas the exchange–correlation effects were included by means of the generalized gradient approximation (GGA) within the Perdew–Burke–Ernzerhof formalism (PBE).[36] Since dispersion interactions play an important role in the accurate prediction of adsorption and desorption energies of hydrocarbon molecules, Grimme's D3 methodology[37] was used to describe the Van der Waals interactions (vdW) semiempirically. At this level of theory, the calculated lattice parameters for the orthorhombic $Mo_2C$ bulk were $a = 4.715$ Å, $b = 5.193$ Å, and $c = 6.035$ Å, which agrees well with the experimental lattice constants ($a = 4.729$ Å, $b = 5.197$ Å, and $c = 6.028$ Å).[38] These parameters were used to generate a ($3 \times 2$) surface unit cell of β-$Mo_2C$(0001) consisting of six atomic layers of $Mo_2C$ and a 15 Å vacuum added in $z$-direction. The cutoff energy for the plane-wave basis was set to 500 eV for the bulk and surface calculations. The reaction mechanism and Cu adsorption were only studied on the Mo-terminated β-$Mo_2C$(0001) surface, similar to previous reports.[15,17,39]

A Cu rod-like structure was created on the β-$Mo_2C$(0001) surface by adding and optimizing three rows of Cu atoms at various positions on the $Mo_2C$ surface to identify the minimum energy configuration shown in the inset of Fig. 4. This structure represents a 0.5 ML Cu on $Mo_2C$ catalyst model, in which 50% of the Mo sites on the $Mo_2C$ surface layer are covered by Cu atoms. This catalyst model allowed us to examine the deoxygenation mechanism of glycerol at three different active sites, namely, the exposed Mo sites on the $Mo_2C$ surface, the Cu sites on the Cu rod, and sites at the Cu-$Mo_2C$ interface. A Monkhorst–Pack (MP)[40] k-point mesh of $2 \times 3 \times 1$ was used, and all atoms were fully relaxed during structural optimization except for the bottom two atomic layers of $Mo_2C$. Dipole and quadrupole corrections to the energy were taken into account using a modified version of the Makov and Payne method[41] and Harris–Foulke-type corrections[42] were included for the forces. The climbing image-NEB[43] and Dimer methods[44,45] were used to optimize all transition state structures.

## Data availability

The data that support the findings of this study are available from the corresponding author upon request.

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

## Acknowledgements

This work was sponsored by the National Science Foundation under Grant No. CHE-1565964. Weiming Wan performed HREELS measurements as part of the Catalysis Center for Energy Innovation (CCEI), an Energy Frontier Research Center (EFRC) funded by the U.S. Department of Energy, Office of Basic Energy Sciences under Award Number DE-SC0001004. Computational resources provided by XSEDE facilities located at San Diego Supercomputer Center (SDSC) and Texas advanced Computing Center (TACC) under grant number TG-CTS090100, U.S. Department of Energy facilities located at the National Energy Research Scientific Computing Center (NERSC) under Contract No. DE-AC02-05CH11231 and Pacific Northwest National Laboratory (Ringgold ID 130367, Grant Proposal 49246) and the High-Performance Computing clusters located at University of South Carolina are gratefully acknowledged.

## Author contributions

W.W. and Z.L. performed surface science experiments. S.C.A. and K.-E.Y. performed DFT calculations. W.W., S.C.A., A.H., and J.G.C. analyzed the results and contributed to writing the paper. A.H. and J.G.C. directed the project. All authors discussed the results and commented on the paper.
