## [Peer Review File · Nature Communications]

Reviewer #1 (Remarks to the Author):

This is a surface science paper involving both theory and experimental measurements. It deals with the modification of a model Mo₂C/Mo(110) catalyst surface with Cu to fine tune the reaction selectivity for the hydrodeoxygenation of glycerol. Taking into account the methods and the chemistry involved in this work I believe the paper is of interest to a relatively broad audience (surface science, catalysis, sustainable chemistry). I believe that this work has the potential to be published in Nature Communications after taking into consideration various points listed below. My main concerns are related to the experimental part of this work:

Comments:

1. Figure 1: I think the desorption spectrum of the reactant (glycerol) should be included. This can be done by following the appropriate mass fragment on the mass spectrometer (e.g. m/z 61 would be good enough to distinguish glycerol from the other molecules).
2. Figure 1(b) (and relevant discussion in the paper): m/z 43 is also a major fragment of glycerol (see NIST webbook). The author should provide further information on how they have distinguished between the glycerol and acetol by showing the relevant data in the supplementary information.
3. Figure 1(c) and 1(d) m/z 57 and m/z 58 are ionization fragments of both propanal and allyl alcohol. Similarly to my previous comment, I am not sure how the authors distinguish between these two products (they both seem to desorb at relatively close temperatures –there seems to be an overlap in the desorption traces). Have the authors fitted and deconvoluted the spectra? It would be very useful if they provide the relevant information.
4. Figure 1(c) and 1(d): what is the peak at ~220K for the MoC₂ and 0.1 ML spectra?
5. The authors deposited Cu on the surface at 300K and monitored the deposition with AES. Taking into account that the chemistry reported in this paper takes place above 300K, the stability of the Cu layer above 300K is very important. Have the authors performed any experiments to address this point (for instance perform AES after successive annealing of the sample).

6. The authors should comment on the carbon balance of the reaction (is there any carbon deposition on the surface after the TPD)?

7. I think there is a problem with the fonts in scheme 1 (linewidth of propylene).

Reviewer #2 (Remarks to the Author):

This contribution describes a combined experimental and theoretical study of the interaction of glycerol with copper modified Mo₂C surfaces. Rather than investigating a truly catalytic process, the authors use TPD measurements of glycerol deposited on the various copper modified Mo₂C surfaces. The authors investigate the products formed from TPD and rationalize their findings using density functional theory calculations. While this study is certainly interesting for the surface science and even catalysis community and does constitute a first step towards a deeper understanding of the selectivity of glycerol conversion on the investigated surfaces, it is certainly not relevant enough for a high-impact journal such as Nature Communications. I therefore recommend publication in a more specialized journal. Additionally there are some points the need to be addressed, see below.

(1) The authors show the reactions towards the different products in scheme 1. They write that “atomic oxygen and atomic hydrogen were formed in all reactions, which can recombinatively desorb in the form of water.” While this may be possible for reaction 3, reaction 1 and 2 have the stoichiometries $3O + 2H$ and $2O + 2H$, respectively. Desorption of water will hence only be possible with additional hydrogen. Additionally, the authors would need to provide experiments and calculations showing that these processes are indeed happening.

(2) The authors further write that “..., suggesting that the desorption of acetol and allyl-alcohol was reaction-limited, i. e., the products desorbed from the surface as soon as they were produced.” I would expect both acetol and allyl-alcohol to have rather strong interactions with the surfaces through vdW forces. The authors should provide their adsorption energy and deduct the free energies at the relevant temperatures to substantiate their claim.

(3) I would strongly advise the authors to provide a better way of showing the results in figures 3-5.

Reviewer #3 (Remarks to the Author):

This paper reports interesting behavior of glycerol over a Mo_2C surface that has been coated to various degrees with Cu. The authors report a general shift in reaction selectivity when depositing over layers of less oxophilic Cu over the more oxophilic Mo_2C . While the combination of TPD and HREELS experiments accompanied with rigorous DFT calculations provide a detailed and compelling story, the difference in selectivity between Mo_2C and Cu catalysts is not unexpected or particularly novel. I commend the authors for the detailed discussion pertaining to the potential for various intermediates on the surface as well as the influence of hydrogen coverage on the surface, and the resulting potential impact that this may play on adsorption strength. The combination of the two at intermediate Cu coverage is very interesting, and they do provide reasonable justification for the observed behavior with the Cu-cluster model.

My primary concern with the paper is the possibility that the materials studied are not active catalysts under reaction conditions. The DFT simulations appear to probe to the point of organic reactant desorption but not site regeneration. The authors clearly show that in most catalysts, potentially with the exception of those with a near monolayer of Cu, oxygen desorption to regenerate the active Mo_2C surface is the most energy demanding step. This implies that the actual material under reaction conditions less than 700K (according to Figure S3) may be an oxy-carbide. These results suggest that if the carbide surface studied is indeed the most active surface, the ability of Cu to facilitate hydrogen spillover and subsequent water removal may have a greater influence on reaction rates in a real system than the oxophilic balance discussion described here. If the materials studied in detail here are not active catalyst materials under realistic reaction conditions, I fear that the general impact of this paper not as great.

There are a few ways the authors could probe this behavior experimentally. The most straightforward would be to conduct sequential TPD and possibly HREELS experiments on a surface that has already desorbed the products of interest. This may significantly change the story, as after heating to 500K, for example, and subsequently cooling to 200K and re-introducing glycerol and hydrogen for a second and third turnover. The results may highlight the importance of Cu for regenerating the active site, and shed some light regarding dynamic changes to the nature of the active site in the presence of an oxygen containing reactant. Alternatively, one could deposit various loadings of Cu on a Mo_2C catalyst and study steady state conversions in a flow reactor, although it may be challenging to mimic the same coverage, if the trends are consistent or similar this may help to show that the observed behavior is applicable to a real catalytic system with multiple turnovers.

Point-by-Point Response to Reviewers' Comments

Reviewers' comments:

Reviewer #1 (Remarks to the Author):

This is a surface science paper involving both theory and experimental measurements. It deals with the modification of a model Mo₂C/Mo(110) catalyst surface with Cu to fine tune the reaction selectivity for the hydrodeoxygenation of glycerol. Taking into account the methods and the chemistry involved in this work I believe the paper is of interest to a relatively broad audience (surface science, catalysis, sustainable chemistry). I believe that this work has the potential to be published in Nature Communications after taking into consideration various points listed below. My main concerns are related to the experimental part of this work:

[Response]:

We thank the reviewer for the valuable suggestions and we have modified the manuscript according to the reviewer's comments.

Comments:

1. Figure 1: I think the desorption spectrum of the reactant (glycerol) should be included. This can be done by following the appropriate mass fragment on the mass spectrometer (e.g., m/z 61 would be good enough to distinguish glycerol from the other molecules).

[Response 1.1]:

We have added the spectra of m/z 61 to the Supplementary Figure 2 and changed the order of the three figures.

Supplementary Figure 2. TPD spectra of a) glycerol ($m/z=61$), b) propylene ($m/z=41$) and c) acetol ($m/z=31$) with an exposure of 4 L glycerol on hydrogen pre-dosed Cu/ Mo_2C surfaces with different Cu coverages

We changed the figure order in the text correspondingly and added the following sentence on page 5 “...The main cracking pattern of glycerol is at $m/z = 61$, and the spectra are shown in Supplementary Fig. 2(a)...”.

2. Figure 1(b) (and relevant discussion in the paper): m/z 43 is also a major fragment of glycerol (see NIST webbook). The author should provide further information on how they have distinguished between the glycerol and acetol by showing the relevant data in the supplementary information.

[Response 1.2]:

Based on Response 1.1, the spectra of mass m/z=61 only shows one peak at 259 K. Therefore, the peak at 365 K in the spectra of m/z=43 is not from glycerol.

3. Figure 1(c) and 1(d) m/z 57 and m/z 58 are ionization fragments of both propanal and allyl alcohol. Similarly to my pervious comment, I am not sure how the authors distinguish between these two products (they both seem to desorb at relatively close temperatures –there seems to be an overlap in the desorption traces). Have the authors fitted and deconvoluted the spectra? It would be very useful if they provide the relevant information.

[Response 1.3]:

We added the following equations and procedures in the Supplementary Information on page 20:

“The quantification results of propylene, acetol, allyl alcohol and propanal were calculated by the following equations:

$$\text{Propylene yield} = \frac{\theta_{H_2}^{sat}}{P_{H_2}^{sat}} P_{propylene}^{39} \frac{S_2^{H_2}}{S_{39}^{propylene}}$$

$$\text{Acetol yield} = \frac{\theta_{H_2}^{sat}}{P_{H_2}^{sat}} P_{acetol}^{43} \frac{S_2^{H_2}}{S_{43}^{acetol}}$$

$$\text{Allyl alcohol yield} = \frac{\theta_{H_2}^{sat}}{P_{H_2}^{sat}} P_{allyl alcohol}^{57} \frac{S_2^{H_2}}{S_{57}^{allyl alcohol}}$$

$$\text{Propanal yield} = \frac{\theta_{H_2}^{sat}}{P_{H_2}^{sat}} P_{propanal}^{58} \frac{S_2^{H_2}}{S_{58}^{propanal}}$$

$\theta_{H_2}^{sat}$ is the saturation coverage of H_2 on the Mo_2C surface and $P_{H_2}^{sat}$ is the TPD area of the 2 amu peak from saturation H_2 desorption. $S_2^{H_2}$, $S_{39}^{propylene}$, S_{43}^{acetol} , $S_{58}^{propanal}$ and $S_{57}^{allyl alcohol}$ are mass spectrometer sensitivities for H_2 , propylene, acetol, propanal and allyl alcohol, respectively. $P_{propylene}^{39}$ and P_{acetol}^{43} were the areas of 39 amu and 43 amu peaks.

The intensity ratio of the two fragments (m/z 57, m/z 58) from allyl alcohol and propanal was measured by Mass-Spec. Based on the calibration result, the intensity ratio of m/z 58 over m/z 57 is 0.176 for allyl alcohol and 3.07 for propanal. Therefore, $P_{Propanal}^{58}$ and $P_{allyl alcohol}^{57}$ were deconvoluted using following equations:

$$P_{allyl alcohol}^{57} + P_{Propanal}^{58} * \frac{1}{3.07} = P^{57}$$

$$P_{\text{Propanal}}^{58} + P_{\text{allyl alcohol}}^{57} * 0.176 = P^{58}$$

P⁵⁷ and P⁵⁸ were the areas of the 57 amu peak and the 58 amu peak from glycerol TPD experiments.”

4. Figure 1(c) and 1(d): what is the peak at ~220K for the MoC₂ and 0.1 ML spectra?

[Response 1.4:]

Based on our experience, the peak at 220 K should be from the heating wires that were connected to the Mo(110) single crystal. The temperature of the heating wires increased rapidly during the initial TPD measurements, leading to the desorption and/or reaction of molecules accumulated on the heating wires during the sample preparation. These peaks were typically relatively weak and occurred before the desorption of glycerol from the Mo₂C/Mo(110) surface.

5. The authors deposited Cu on the surface at 300K and monitored the deposition with AES. Taking into account that the chemistry reported in this paper takes place above 300K, the stability of the Cu layer above 300K is very important. Have the authors performed any experiments to address this point (for instance perform AES after successive annealing of the sample).

[Response 1.5:]

To verify the stability of the Cu layer, we deposited 1 ML Cu on the Mo₂C surface at 300 K and heated the surface at 3 K/s to increasingly high temperatures. The AES ratio of Cu(64eV)/Mo(186eV) was plotted as a function of the annealing temperature and was added to the Supplementary Figure 1(b). The result shows that the AES ratio is within 10% of the initial value at temperatures below 600 K. Since the HDO products desorbed below 600 K, the change of Cu structures (agglomeration or diffusion) should not affect the selective HDO reaction.

To clarify this, we added the following sentence on page 18, *“The Supplementary Fig. 1(b) shows the thermal stability of ~ 1 ML Cu on Mo₂C surface. The Cu/Mo AES ratio is within 10% of the initial value at temperatures below 600 K, suggesting that the Cu overlayer remains relatively stable in the temperature range of the HDO reactions.”*

Supplementary Figure 1. a) AES measurements of Cu deposition on Mo₂C/Mo(110) at 300 K as a function of deposition time, b) AES measurements of Cu(64 eV)/Mo(186 eV) ratio for 1 ML Cu/ Mo₂C/Mo(110) as a function of annealing temperature

6. The authors should comment on the carbon balance of the reaction (is there any carbon deposition on the surface after the TPD)?

[Response 1.6:]

We used the atomic ratio of C(275 eV)/Mo(186 eV) to determine the amount of surface carbon. The AES measurements were performed before and after each TPD experiment. The C/Mo ratio increase less than 1% after each TPD experiment, indicating that only a trace amount of carbon was deposited on the surface.

We added the following sentence on page 4, “*Only trace amount of atomic carbon was deposited after each TPD experiment, which was observed in AES measurements.*”

7. I think there is a problem with the fonts in scheme 1 (linewidth of propylene).

[Response 1.7:]

We have modified the fonts in scheme 1 as follows:

Scheme 1. Reaction pathways for the selective hydrodeoxygenation of glycerol

Reviewer #2 (Remarks to the Author):

This contribution describes a combined experimental and theoretical study of the interaction of glycerol with copper modified Mo₂C surfaces. Rather than investigating a truly catalytic process, the authors use TPD measurements of glycerol deposited on the various copper modified Mo₂C surfaces. The authors investigate the products formed from TPD and rationalize their findings using density functional theory calculations. While this study is certainly interesting for the surface science and even catalysis community and does constitute a first step towards a deeper understanding of the selectivity of glycerol conversion on the investigated surfaces, it is certainly not relevant enough for a high-impact journal such as Nature Communications.

I, therefore, recommend publication in a more specialized journal. Additionally there are some points the need to be addressed, see below.

[Response:]

We thank the reviewer for the valuable suggestions, and we have modified the manuscript according to the reviewer's comments. Regarding the novelty of the paper, to the best of our knowledge, this paper represents the first attempt to combine DFT calculations of reaction networks of glycerol with experimental determination of surface intermediates over model surfaces. Because DFT calculations are typically performed on single crystal surfaces, it is often difficult to compare DFT predictions with experimental results on heterogeneous supported catalysts. By using HREELS and TPD to follow the surface intermediates and reaction products on single crystal Cu/Mo₂C surfaces and interfaces, our paper allows the direct and meaningful comparison between theory and experiments for the HDO reactions of glycerol. The identification of surface active sites and bonding configurations in the current study should provide guidance in designing catalyst compositions and structures for the selective upgrading of glycerol, which has not been well studied in terms of fundamental reaction pathways despite being one of the most important biomass-derived platform molecules.

(1) The authors show the reactions towards the different products in scheme 1. They write that "atomic oxygen and atomic hydrogen were formed in all reactions, which can recombinatively desorb in the form of water." While this may be possible for reaction 3, reaction 1 and 2 have the stoichiometries 3O + 2H and 2O + 2H, respectively. Desorption of water will hence only be possible with additional hydrogen. Additionally, the authors would need to provide experiments and calculations showing that these processes are indeed happening.

[Response 2.1:]

We thank the reviewer for pointing out the stoichiometric issue for the oxygen removal. In all TPD experiments, 4 Langmuir hydrogen was pre-dosed on the surface, which provided extra atomic hydrogen to remove the surface oxygen. To clarify this, we added the following sentence on page 4: "In all TPD experiments, 4 Langmuir hydrogen was pre-dosed on the surface, which provided extra atomic hydrogen to remove the surface oxygen."

We also thank the reviewer for the concerns about the mechanism for the removal of surface oxygen. To examine the pathways for the removal of adsorbed oxygen atoms from the three

different active sites, we performed additional DFT calculations. The reaction energies and activation barriers of each possible elemental step were summarized in the Supplementary Table 1.

Based on the DFT calculations, on the Mo₂C surface H₂O formation was favorable by the disproportionation reaction of two neighboring -OH groups (reaction 4) rather than the transfer of adsorbed H to the -OH groups (reaction 2). The surface oxygen preferred to be hydrogenated by gas-phase H₂ (reaction 3) rather than atomic hydrogen (reaction 1). However, since the TPD experiments were performed under UHV condition, gas phase H₂ was not involved. Therefore, the surface oxygen had to be hydrogenated by atomic hydrogen, forming -OH groups, which produced water through the disproportionation reaction. In this process, the highest activation barrier for the water formation on Mo₂C surface was 2.08 eV.

On the Cu-Mo₂C interface, the water formation also preferred the disproportionation of two neighboring -OH groups on the Mo sites (reaction 10). Due to the Cu modifier, the activation barrier for the oxygen hydrogenation was much lower at the interface (reaction 7). On the Cu-Mo₂C interface, the highest activation barrier for water formation was 0.92 eV. Similarly, on the Cu surface, the highest barrier for water formation was 1.05 eV.

Overall, the DFT results show that water formation has a high barrier on the Mo₂C surface and lower barriers on the Cu-Mo₂C interface and the Cu surface. Based on the TPD experiments (Supplementary Figure 3), the water desorption peak was at 569 K on the Mo₂C surface and was at 365 K on the Cu-Mo₂C interface and the Cu surface. The higher water desorption temperature from the Mo₂C surface is consistent with the higher activation barrier of water formation from the DFT calculations.

Supplementary Table 1. Zero point energy corrected reaction energies (ΔE_{ZPE}) and activation barriers (E_{ZPE}^{act}) calculated for the formation of H₂O on different active sites of the Cu/Mo₂C (0001) catalyst model

Reaction	ΔE_{ZPE} (eV)	E_{ZPE}^{act} (eV)
Mo sites of the Cu/Mo₂C catalyst		
(1) O* + H* → OH* + *	1.20	2.08
(2) OH* + H* → H ₂ O* + *	1.52	1.90
(3) O* + H ₂ (g) → OH* + H*	-0.56	0.38
(4) OH* + OH* → H ₂ O* + O*	0.42	0.75
(5) H ₂ O* → H ₂ O(g) + *	0.94	0.94
Cu/Mo₂C interface sites		
(6) H ₂ (g) + 2* _{Cu} → 2H _{Cu} *	-0.51	0.17
(7) 2H _{Cu} * + O* _{Mo} → OH _{Mo} * + H _{Cu} * + * _{Cu}	0.47	0.92
(8) OH _{Mo} * + H _{Cu} * → H ₂ O _{Mo} * + * _{Cu}	1.03	1.12
(9) O _{Mo} * + H ₂ (g) → OH _{Mo} * + H _{Mo} *	-0.67	0.39
(10) OH _{Mo} * + OH _{Mo} * → H ₂ O _{Mo} * + O _{Mo} *	0.44	0.61

(11) $\text{H}_2\text{O}_{\text{Mo}}^* \rightarrow \text{H}_2\text{O}(\text{g}) + ^*\text{Mo}$	0.82	0.82
Cu sites of the Cu/Mo₂C catalyst		
(12) $\text{O}^* + \text{H}^* \rightarrow \text{OH}^* + ^*$	-0.48	1.05
(13) $\text{OH}^* + \text{H}^* \rightarrow \text{H}_2\text{O}^* + ^*$	0.79	1.36
(14) $\text{O}^* + \text{H}_2(\text{g}) \rightarrow \text{OH}^* + \text{H}^*$	-0.81	0.33
(15) $\text{OH}^* + \text{OH}^* \rightarrow \text{H}_2\text{O}^* + \text{O}^*$	0.42	0.44
(16) $\text{H}_2\text{O}^* \rightarrow \text{H}_2\text{O}(\text{g}) + ^*$	0.35	0.35

Supplementary Figure 3. TPD spectra of water ($m/z=18$) with an exposure of 4 L glycerol on hydrogen pre-dosed Cu/Mo₂C surfaces with different Cu coverages

To clarify this, we added the following sentences on page 6 and page 7:” *To confirm the surface oxygen was removed by the surface hydrogen, DFT calculations were performed to compare the largest activation barriers for oxygen removal from the Mo₂C surface, the Cu-Mo₂C interface and the Cu surface. As shown in Supplementary Table 1, under the UHV condition, gas phase H₂ did not participate in the oxygen removal, and the surface oxygen was removed by atomic hydrogen. The highest activation barrier is 2.08 eV, 0.94 eV and 1.05 eV on the three surfaces, respectively.*

In the TPD experiments (Supplementary Fig. 3), the water desorption peak was at 569 K on the Mo₂C surface, and at ~365 K on the Cu-Mo₂C interface and the Cu surface. The higher water desorption temperature from the Mo₂C surface is consistent with the higher activation barrier of water formation from the DFT calculations.”

(2) The authors further write that “...., suggesting that the desorption of acetol and allyl-alcohol was reaction-limited, i. e., the products desorbed from the surface as soon as they were produced.” I would expect both acetol and allyl-alcohol to have rather strong interactions with the surfaces through vdW forces. The authors should provide their adsorption energy and deduct the free energies at the relevant temperatures to substantiate their claim.

[Response 2.2:]

Our calculations suggest that acetol and allyl alcohol are produced on the Cu surface and Cu/Mo₂C interface sites, respectively, and thus, the final products are adsorbed on the Cu sites. The ZPE corrected adsorption energies for these products, and the corresponding free energies are shown in the Supplementary Tables 5 and 6 and Supplementary Fig. 12. The calculated adsorption energy of acetol is -0.84 eV (state (6) → (7), Supplementary Table 6) and that of allyl alcohol is -1.38 eV (state (5A) → (6A), Supplementary Table 5). The free energy diagram indicates that desorption of these products is exergonic at a temperature of 350 K, i.e., facile desorption.

(3) I would strongly advise the authors to provide a better way of showing the results in figures 3-5.

[Response 2.3:]

We agree with the reviewer and modified Figures 3-5 with stick diagrams. We believe that the new figures provide a more clear description of the reaction pathways.

Reviewer #3 (Remarks to the Author):

This paper reports the interesting behavior of glycerol over a Mo₂C surface that has been coated to various degrees with Cu. The authors report a general shift in reaction selectivity when depositing over layers of less oxophilic Cu over the more oxophilic Mo₂C. While the combination of TPD and HREELS experiments accompanied with rigorous DFT calculations provide a detailed and compelling story, the difference in selectivity between Mo₂C and Cu catalysts is not unexpected or particularly novel.

[Response:]

We thank the reviewer for the valuable suggestions, and we have modified the manuscript according to the reviewer's comments. Regarding the novelty of the paper, to the best of our knowledge, this paper represents the first attempt to combine DFT calculations of reaction networks of glycerol with experimental determination of surface intermediates over model surfaces. Because DFT calculations are typically performed on single crystal surfaces, it is often difficult to compare DFT predictions with experimental results on heterogeneous supported catalysts. By using HREELS and TPD to follow the surface intermediates and reaction products on single crystal Cu/Mo₂C surfaces and interfaces, our paper allows the direct and meaningful comparison between theory and experiments for the HDO reactions of glycerol. The identification of surface active sites and bonding configurations in the current study should provide guidance in designing catalyst compositions and structures for the selective upgrading of glycerol, which has not been well studied in terms of fundamental reaction pathways despite being one of the most important biomass-derived platform molecules.

1. I commend the authors for the detailed discussion pertaining to the potential for various intermediates on the surface as well as the influence of hydrogen coverage on the surface and the resulting potential impact that this may play on adsorption strength. The combination of the two at intermediate Cu coverage is very interesting, and they do provide reasonable justification for the observed behavior with the Cu-cluster model.

[Response 3.1:]

We performed TPD experiments on Mo₂C, 0.5 ML Cu/Mo₂C and 1 ML Cu/Mo₂C surfaces. On each surface 0 L, 5 L and 100 L H₂ were pre-dosed to study the hydrogen coverage effect. The spectrum of m/z=39 on the Mo₂C surface reveals propylene formation on the Mo₂C site (Supplementary Figure 4(a)), the spectrum of m/z=43 on the 1 ML Cu/Mo₂C surface demonstrates the acetol formation on the Cu site (Supplementary Figure 4(b)), and the spectrum of m/z=57,58 on the 0.5 ML Cu/Mo₂C surface suggests the allyl-alcohol/propanal formation on the Cu-Mo₂C interface (Supplementary Figure 4(c), Supplementary Figure 4(d)).

As the amount of pre-dosed H₂ increased from 0 L to 100 L, the peak temperatures and the areas of the three desorption peaks in all spectra did not have a significant change, suggesting that the hydrogen coverage did not have a significant effect on the three reactions: propylene formation (Supplementary Figure 4(a)), acetol formation (Supplementary Figure 4(b)) and allyl-alcohol/propanal formation (Supplementary Figure 4(c), Supplementary Figure 4(d)).

To clarify this, the following sentence was added to page 7:” *The hydrogen coverage effect was studied (Supplementary Fig. 4) and the pre-dosed amount did not affect the activities of the three distinguish sites.*”

For the potential intermediates, due to the lack of pure chemical samples, it is not possible to perform the TPD by directly dosing possible intermediates such as dihydroxyacetone and glyceraldehyde. Therefore, we infer the reaction intermediates based on the experimentally measured product distribution and DFT calculations.

Supplementary Figure 4. TPD spectra for: a) propylene desorption on Mo₂C, b) acetol desorption on 1 ML Cu/Mo₂C, c) allyl-alcohol, and d) propanal desorption on 0.5 ML Cu/Mo₂C with 0 L, 5 L and 100 L H₂ pre-dosed

2. My primary concern with the paper is the possibility that the materials studied are not active catalysts under reaction conditions. The DFT simulations appear to probe to the point of organic reactant desorption but not site regeneration. The authors clearly show that in most catalysts, potentially with the exception of those with a near monolayer of Cu, oxygen desorption to regenerate the active Mo_2C surface is the most energy demanding step. This implies that the actual material under reaction conditions less than 700K (according to Figure 3) may be an oxy-carbide. These results suggest that if the carbide surface studied is indeed the most active surface, the ability of Cu to facilitate hydrogen spillover and subsequent water removal may have a greater influence on reaction rates in a real system than the oxophilic balance discussion described here. If the materials studied in detail here are not active catalyst materials under realistic reaction conditions, I fear that the general impact of this paper not as great. There are a few ways the authors could probe this behavior experimentally. The most straightforward would be to conduct sequential TPD and possibly HREELS experiments on a surface that has already desorbed the products of interest. This may significantly change the story, as after heating to 500K, for example, and subsequently cooling to 200K and re-introducing glycerol and hydrogen for a second and third turnover. The results may highlight the importance of Cu for regenerating the active site, and shed some light regarding dynamic changes to the nature of the active site in the presence of an oxygen containing reactant.

[Response 3.2:]

To study the role of Cu modifier in catalyst regeneration, we performed sequential TPD experiments on three surfaces, Mo_2C , 0.5 ML Cu/ Mo_2C and 1 ML Cu/ Mo_2C , between 200 K and 500 K. Figure R1(a) shows the desorption of propylene on the Mo_2C surface. In the second TPD experiment, the propylene desorption peak shifts to 470 K and the intensity reduces, suggesting a lower C-O bond cleavage activity of the Mo_2C surface after the first TPD experiment. In contrast, on the 1 ML Cu/ Mo_2C surface (Figure R1(b)), the Cu site maintains activity in three sequential TPD experiments. In Figure R1(c), the 0.5 ML Cu/ Mo_2C also loses activity towards allyl alcohol formation after the first TPD experiment. In the second TPD experiment, a small peak of propanal was observed at 402 K (Figure R1(d)). The TPD results under the UHV condition suggest that the Mo_2C surface and Cu- Mo_2C interface are less stable than the Cu surface, which is due to the strong oxophilicity of the Mo site.

However, these experiments were performed under the UHV condition, in which the gas-phase H_2 was not involved. In a real catalytic HDO process with a high H_2 pressure, the gas phase H_2 should help remove surface oxygen and maintain a higher hydrogen coverage. In previous research, Ren *et al.* studied the HDO reaction of propanal on powder Mo_2C catalyst using a flow reactor.¹ The catalyst quickly deactivated in the absence of H_2 . With co-feed H_2 , a steady-state propanal conversion of approximately 50% was achieved at 573 K, suggesting that oxygen was removed by H_2 . This was also confirmed by the production of water at this temperature. These results are also consistent with DFT results of lower activation barriers for oxygen removal in the presence of gas-phase H_2 , as shown in Supplementary Table 1.

Figure R1. Sequential TPD results for: a) propylene desorption on Mo₂C, b) acetol desorption on 1 ML Cu/Mo₂C, c) allyl-alcohol, and d) propanal desorption on 0.5 ML Cu/Mo₂C

In order to further substantiate the hypothesis that the adsorbed oxygen can be removed from the “oxycarbide” surface under reaction conditions, we performed DFT calculations on the model surfaces used in the present study, in which all exposed Mo sites are occupied by oxygen atoms. For this model, the oxygen vacancy formation free energy was found to range from 0.4 - 1.2 eV ($O^* + H_2 \rightarrow O_{\text{vac}} + H_2O$, $T = 500 \text{ K}$; $P_{\text{gas}}=1 \text{ atm}$) when going from the Cu/Mo₂C interface site to the sites away from the Cu. Thus, we examined the possibility of H spillover from Cu to the interface oxygens to form H₂O (Supplementary Fig. 15). The calculated barriers were found to be around 1 eV, and our calculations suggest that the removal of interface oxygen is feasible at 500 K when P_{H_2}/P_{H_2O} is above 10^2 .

These results are added in the Supplementary Information on page 16-19: *In order to further examine the removal of adsorbed oxygen atoms from the three different active sites, activation barriers were calculated for the H₂O formation process, both in the presence and absence of excess H₂ (see Supplementary Table 1). On the Mo sites, hydrogenation of either O* or OH* with adsorbed H* was found to be highly endothermic with barriers of about 2 eV (reactions 1 & 2). However, direct dissociation of gas phase H₂ onto O* was found to be favorable with an activation barrier of only 0.38 eV. Similarly, the formation of H₂O via the disproportionation reaction of two neighboring OH* was also found to possess a barrier of only 0.75 eV. These results are consistent with the earlier report by Ren et al. that the adsorbed O on the Mo sites can be removed by excess H₂.¹ On the Cu/Mo₂C interface sites, we examined the possibility of H₂ dissociation on the Cu sites and two subsequent H atom spillovers from the Cu to the interface oxygen adsorbed on the Mo sites to form H₂O. Calculations revealed that the dissociation of gas phase H₂ on Cu is an exothermic process and the spillover of H atoms from Cu to interface O requires overcoming barriers of only about 1 eV, suggesting that these processes are feasible. The direct dissociation of gas phase H₂ onto O* (reaction 9) and OH disproportionation (reaction 10) were also examined at the interface site for which the barriers were calculated as 0.39 and 0.61 eV, respectively. H₂O formation is thus favorable both on the Mo sites and at the interface sites via OH disproportionation provided that empty Mo sites are available at neighboring sites that can promote the formation of OH groups by direct dissociation of gas phase H₂ onto adsorbed O.*

On the Cu sites, the activation barriers calculated for H₂O formation in the absence of excess H₂ was found to be >1 eV (reactions 12 & 13 in Supplementary Table 1). However, these barriers are smaller than the corresponding barriers on the Mo sites. The calculated barriers for the stepwise hydrogenation of O (1.05 eV) and OH* (1.36 eV) are similar to those reported for different terminations of Cu surfaces^{4,5}. Here again, direct dissociation of excess gas phase H₂ onto adsorbed O* (reaction 14) and the OH disproportionation to form H₂O (reaction 15) were found to be more favorable than the stepwise hydrogenation processes. The endothermicity of H₂O desorption from the three sites decreases in the order, Mo site (0.94 eV) > interface site (0.82 eV) > Cu site (0.35 eV), which is consistent with the decrease in the desorption temperature observed in the TPD spectra for H₂O desorption when going from pure Mo₂C to 1 ML Cu/Mo₂C (Supplementary Fig. 3).*

These results suggest that H₂O formation from adsorbed O on the three surfaces are possible in the presence of excess H₂ with activation barriers of less than 1 eV. Since the partial pressure of H₂O at UHV conditions is as low as 10^{-12} atm , H₂O can easily desorb from these sites. However, under experimental reactor conditions, i.e., ambient pressure conditions, all the Mo sites could be occupied by oxygen atoms. In fact, constrained ab initio thermodynamic analysis

carried out in the presence of an equimolar H_2/H_2O gas phase for our current catalyst model suggested that all the exposed Mo sites are occupied by oxygen atoms at a temperature of 500 K. For this model (1st structure in Supplementary Fig. 15), the oxygen vacancy formation free energy was found to range from 0.4 to 1.2 eV ($O^* + H_2 \rightarrow O_{vac} + H_2O$, $T = 500$ K; $P_{gas}=1$ atm) when going from the Cu/Mo₂C interface site to the sites away from Cu. Thus, we examined the possibility of H spillover from Cu to the interface oxygens to form H₂O. The calculated barriers for the 1st and 2nd H spillover process were found to be similar but slightly more favorable on the fully oxygen covered model compared to those calculated for the model with lower oxygen coverage (Supplementary Table 1). The free energy profile calculated at a temperature of 500 K suggested that the removal of interface oxygen is feasible when the P_{H_2}/P_{H_2O} is slightly above 10^2 which can easily be achieved experimentally. The effective free energy barrier of 1.36 eV corresponds to an approximate oxygen removal rate of $10^{-1} s^{-1}$, indicating that the interface oxygens can be removed from an “oxycarbide” surface and the Cu/Mo₂C catalyst should remain active at more practical reactor conditions.

Supplementary Figure 15. Reaction pathways calculated for the removal of interface oxygen from the oxygen covered Cu/Mo₂C(0001) catalyst model. Bond distances shown next to the breaking/forming bonds are in Å

Supplementary Table 1. Zero point energy corrected reaction energies (ΔE_{ZPE}) and activation barriers (E_{ZPE}^{act}) calculated for the formation of H₂O on different active sites of the Cu/Mo₂C(0001) catalyst model

Reaction	ΔE_{ZPE} (eV)	E_{ZPE}^{act} (eV)
Mo sites of the Cu/Mo₂C catalyst		
(1) O* + H* → OH* + *	1.20	2.08

(2) $\text{OH}^* + \text{H}^* \rightarrow \text{H}_2\text{O}^* + *$	1.52	1.90
(3) $\text{O}^* + \text{H}_2(\text{g}) \rightarrow \text{OH}^* + \text{H}^*$	-0.56	0.38
(4) $\text{OH}^* + \text{OH}^* \rightarrow \text{H}_2\text{O}^* + \text{O}^*$	0.42	0.75
(5) $\text{H}_2\text{O}^* \rightarrow \text{H}_2\text{O}(\text{g}) + *$	0.94	0.94
Cu/Mo₂C interface sites		
(6) $\text{H}_2(\text{g}) + 2^*_{\text{Cu}} \rightarrow 2\text{H}_{\text{Cu}}^*$	-0.51	0.17
(7) $2\text{H}_{\text{Cu}}^* + \text{O}^*_{\text{Mo}} \rightarrow \text{OH}_{\text{Mo}}^* + \text{H}_{\text{Cu}}^* + ^*_{\text{Cu}}$	0.47	0.92
(8) $\text{OH}_{\text{Mo}}^* + \text{H}_{\text{Cu}}^* \rightarrow \text{H}_2\text{O}_{\text{Mo}}^* + ^*_{\text{Cu}}$	1.03	1.12
(9) $\text{O}_{\text{Mo}}^* + \text{H}_2(\text{g}) \rightarrow \text{OH}_{\text{Mo}}^* + \text{H}_{\text{Mo}}^*$	-0.67	0.39
(10) $\text{OH}_{\text{Mo}}^* + \text{OH}_{\text{Mo}}^* \rightarrow \text{H}_2\text{O}_{\text{Mo}}^* + \text{O}_{\text{Mo}}^*$	0.44	0.61
(11) $\text{H}_2\text{O}_{\text{Mo}}^* \rightarrow \text{H}_2\text{O}(\text{g}) + ^*_{\text{Mo}}$	0.82	0.82
Cu sites of the Cu/Mo₂C catalyst		
(12) $\text{O}^* + \text{H}^* \rightarrow \text{OH}^* + *$	-0.48	1.05
(13) $\text{OH}^* + \text{H}^* \rightarrow \text{H}_2\text{O}^* + *$	0.79	1.36
(14) $\text{O}^* + \text{H}_2(\text{g}) \rightarrow \text{OH}^* + \text{H}^*$	-0.81	0.33
(15) $\text{OH}^* + \text{OH}^* \rightarrow \text{H}_2\text{O}^* + \text{O}^*$	0.42	0.44
(16) $\text{H}_2\text{O}^* \rightarrow \text{H}_2\text{O}(\text{g}) + *$	0.35	0.35

In the manuscript, the following sentences were added to page 16: “ *Here, we also show computational results that suggest O adatoms deposited on Mo sites in direct neighborhood of Cu sites can be regenerated by H₂ at high oxygen coverage that is likely present under practical reactor conditions (Supplementary Fig. 15)...* ”; and to page 17: “ *...Interestingly, computational results suggest that the O adatoms deposited on Mo sites in direct neighborhood of Cu sites can be regenerated by H₂ at high oxygen coverage likely present under reactor conditions...* ”

3. Alternatively, one could deposit various loadings of Cu on a Mo₂C catalyst and study steady state conversions in a flow reactor, although it may be challenging to mimic the same coverage, if the trends are consistent or similar this may help to show that the observed behavior is applicable to a real catalytic system with multiple turnovers.

[Response 3.3:]

Due to the extremely low vapor pressure of glycerol, it is difficult for us to perform gas-phase flow reactor experiments. Although in principle it should be easier to perform liquid-phase experiments, we do not have the experimental set-up to perform liquid-phase studies. We very much appreciate the critical suggestion by the reviewer, and we hope that our interesting results on model systems would inspire experts in reactor design to perform these difficult experiments.

Reference

1. Ren, H., Yu, W., Saliccioli, M., Chen, Y., Huang, Y., Xiong, K., Vlachos, D. G. & Chen, J. G. Selective hydrodeoxygenation of biomass-derived oxygenates to unsaturated hydrocarbons using molybdenum carbide catalysts. *ChemSusChem* **6**, 798–801 (2013).

Reviewer #1 (Remarks to the Author):

The authors have addressed carefully the comments raised by all referees. Substantial new information has been incorporated into the manuscript (especially with respect to TPDs and also AES data). This new information has improved substantially the manuscript which in my view is ready for publication in Nature Communications.

Reviewer #2 (Remarks to the Author):

The reviewers addressed my concerns regarding the scientific issues of the manuscript.

I still do not see the high relevance needed for publication in Nature Communications, though.

As the authors write, this study represents the first attempt to combine theory and experiments on single crystal surfaces for glycerol desorption and reaction. While this is certainly interesting from a mechanistic point of view, the TPD of glycerol deposited on single crystal surfaces is far from realistic catalytic conditions of HDO. How much one can learn for real applications from this surface science study thus remains to be seen. Furthermore, the decomposition of glycerol has been the subject of theoretical studies, see e.g. reference 25.

Reviewer #3 (Remarks to the Author):

The authors have gone to great lengths to improve this manuscript and address my primary concern, that this material may not be the active catalyst surface under reaction conditions. While the presented DFT results are very helpful, they present a case where clean carbide sites may exist under some cases at higher hydrogen pressures. The direct interaction of gas phase hydrogen implies that under the significant hydrogen pressures, the shifts in TP peaks may be quite different if surface oxygen was readily removed. Clearly after one cycle, the surface is becoming oxidized, and it is not clear to what degree the oxygen that has not been removed from the surface influences the resulting evolution temperature and product selectivity of the resulting TP peaks. In a real flowing system, the coverage of oxygen will vary with both glycerol and hydrogen partial pressure, and possibly create partially oxidized sites with very different binding energies. While a simple TPR with higher hydrogen pressures would be much more convincing, to show at what temperature the surface can be regenerated, I feel that the discussion that the authors have added in the supplemental information is helpful. It unfortunately does not tell an entirely complete story. For transparency purposes, I suggest that the authors include Figure R1 in the supporting information

and refer to it in the manuscript. This demonstrates that the surface does indeed change during the course of the initial TPD experiment, and this is important information for the reader to have access to.

If this paper is accepted, it will likely be read and cited by many. Because of this, it is even more important to highlight the possibility that the sites studied here may or may not be present in significant quantities under actual reaction conditions. Adding these details should help avoid confusion that could arise in the future.

Response to Reviewers' Comments

Reviewer #2 (Remarks to the Author):

The reviewers addressed my concerns regarding the scientific issues of the manuscript.

I still do not see the high relevance needed for publication in Nature Communications, though.

As the authors write, this study represents the first attempt to combine theory and experiments on single crystal surfaces for glycerol desorption and reaction. While this is certainly interesting from a mechanistic point of view, the TPD of glycerol deposited on single crystal surfaces is far from realistic catalytic conditions of HDO. How much one can learn for real applications from this surface science study thus remains to be seen. Furthermore, the decomposition of glycerol has been the subject of theoretical studies, see e.g. reference 25.

[Response]:

We respectfully disagree with the Reviewer's concern regarding the relevance between TPD on single crystal surfaces and realistic catalytic conditions. Our combined experimental and DFT efforts on single crystal surfaces conclusively identified the active sites responsible for the three distinct reaction pathways of glycerol. With the tremendous advances in catalyst synthesis and in-situ characterization, our results should provide guidance to catalysis researchers to optimize their synthesis methods and reaction conditions to potentially achieve the desired active sites. This is especially important for the selective transformations of an important and plentiful platform molecule such as glycerol.

Reviewer #3 (Remarks to the Author):

The authors have gone to great lengths to improve this manuscript and address my primary concern, that this material may not be the active catalyst surface under reaction conditions. While the presented DFT results are very helpful, they present a case where clean carbide sites may exist under some cases at higher hydrogen pressures. The direct interaction of gas phase hydrogen implies that under the significant hydrogen pressures, the shifts in TP peaks may be quite different if surface oxygen was readily removed. Clearly after one cycle, the surface is becoming oxidized, and it is not clear to what degree the oxygen that has not been removed from the surface influences the resulting evolution temperature and product selectivity of the resulting TP peaks. In a real flowing system, the coverage of oxygen will vary with both glycerol and hydrogen partial pressure, and possibly create partially oxidized sites with very different binding energies. While a simple TPR with higher hydrogen pressures would be much more convincing, to show at what temperature the surface can be regenerated, I feel that the discussion that the authors have added in the supplemental information is helpful. It unfortunately does not tell an entirely complete story. For transparency purposes, I suggest that the authors include Figure R1 in the supporting information and refer to it in the manuscript. This demonstrates that the surface does indeed change during the course of the initial TPD experiment, and this is important information for the reader to have access to.

If this paper is accepted, it will likely be read and cited by many. Because of this, it is even more important to highlight the possibility that the sites studied here may or may not be present in significant quantities under actual reaction conditions. Adding these details should help avoid confusion that could arise in the future.

[Response]:

We thank the reviewer for the valuable suggestions, and we have moved Figure R1 to the Supplementary Figure 5 and added the following sentences on Page 7 of the main text: “*To study the role of Cu modifier in catalyst regeneration, sequential TPD experiments were performed under UHV conditions (Supplementary Fig. 5), suggesting that the Mo₂C surface and Cu-Mo₂C interface are less stable than the Cu surface. This is related to the strong oxophilicity of the Mo site that prevents oxygen removal under UHV conditions. However, in a real catalytic HDO process with a high H₂ pressure the gas phase H₂ should help remove surface oxygen (Supplementary Discussion), as observed in a previous study.^{17,}*”

We also added the following statement to the supplementary information on Page 24~25:”

Surface regeneration under UHV condition

To study the role of Cu modifier in catalyst regeneration, sequential TPD experiments were performed on three surfaces, Mo₂C, 0.5 ML Cu/Mo₂C and 1 ML Cu/Mo₂C, between 200 K and 500 K. Supplementary Fig. 5(a) shows the desorption of propylene on the Mo₂C surface. In the second TPD experiment, the propylene desorption peak shifts to 470 K and the intensity reduces, suggesting a lower C-O bond cleavage activity of the Mo₂C surface after the first TPD experiment. In contrast, on the 1 ML Cu/Mo₂C surface (Supplementary Fig. 5(b)), the Cu site maintains activity in three sequential TPD experiments. In Supplementary Fig. 5(c), the 0.5 ML Cu/Mo₂C also loses activity toward allyl alcohol formation after the first TPD experiment. In the second TPD experiment, a small peak of propanal is observed at 402 K (Supplementary Fig. 5(d)). The TPD results under the UHV condition suggest that the Mo₂C surface and Cu-Mo₂C interface are less stable than the Cu surface, which is due to the strong oxophilicity of the Mo site.

However, these experiments were performed under UHV conditions, in which gas-phase H₂ is not involved. In a real catalytic HDO process with a high H₂ pressure, gas-phase H₂ should help remove surface oxygen and maintain a higher hydrogen coverage. In a previous study, Ren et al. studied the HDO reaction of propanal on powder Mo₂C catalyst using a flow reactor.³ The catalyst quickly deactivated in the absence of H₂. With co-feed H₂, a steady-state propanal conversion of approximately 50% was achieved at 573 K, suggesting that oxygen was removed by H₂. This was also confirmed by the production of water at this temperature. These results are also consistent with DFT results in the current study of lower activation barriers for oxygen removal in the presence of gas-phase H₂ (Supplementary Table 1).”